# Gut Microbiota, Intestinal Morphometric Characteristics, and Gene Expression in Relation to the Growth Performance of Chickens

**DOI:** 10.3390/ani12243474

**Published:** 2022-12-09

**Authors:** El-Sayed M. Abdel-Kafy, Sabbah F. Youssef, Mahmoud Magdy, Shereen S. Ghoneim, Hesham A. Abdelatif, Randa A. Deif-Allah, Yasmein Z. Abdel-Ghafar, Hoda M. A. Shabaan, Huazhen Liu, Abdelmotaleb Elokil

**Affiliations:** 1Animal Production Research Institute (APRI), Agricultural Research Center (ARC), Dokki, Giza 12651, Egypt; 2Genetics Department, Faculty of Agriculture, Ain Shams University, Cairo 11241, Egypt; 3Department of Basic Veterinary Medicine, College of Animal Science and Veterinary Medicine, Huazhong Agricultural University, Wuhan 430070, China; 4Department of Animal Production, Faculty of Agriculture, Benha University, Moshtohor 13736, Egypt

**Keywords:** chickens, microbiota, gastrointestinal, histological characteristics, gene expression, blood biochemical

## Abstract

**Simple Summary:**

this study aimed to investigate the growth of chickens by comparing their microbiota, histological characteristics, and gene expression in a local chicken breed. Four hundred and eighty Golden Montazah chickens, an Egyptian breed, were reared until they were 49 days old, and the forty-eight birds with the heaviest body weight (HW) and the lightest body weight (LW) were chosen. The positive histological changes increased significantly in the HW chickens compared to the LW chickens. The growth of the chickens may be affected by intestinal microbiota, and it has a role in feed conversion efficiency. Our findings showed that there is a difference in the microbial community colonized in the gut between the high-weight and low-weight birds which may improve the intestinal functions.

**Abstract:**

this study aimed to investigate the growth mechanism in a local breed of chickens by comparing the highest weight (HW) and the lowest weight in their microbiota, histological characteristics, and gene expression. Golden Montazah chickens, an Egyptian breed, were reared until they were 49 days old. All of the birds were fed ad libitum by a starter diet from day 1 until day 21, followed by a grower diet from day 21 to the end of the study. At 49 days old, the forty-eight birds with the heaviest body weight (HW) and the lightest body weight (LW) were chosen. Blood biochemical and histological morphometric parameters, electron microscopy, and intestinal nutrient transporter gene expression were studied in the sampled jejunum. The microbial composition and functions of the content and mucosa in HW and LW chickens were studied using 16S rRNA gene sequencing. The histological morphometric parameters were all more significantly (*p* < 0.05) increased in the HW chickens than in the LW chickens. Total protein, albumin, and triglycerides in serum were significantly higher (*p* < 0.05) in the HW chickens than in the LW chickens. The microbiome profile in the gut showed that Microbacterium and Sphingomonas were positively correlated with the body weights. In the local breed, there were significant differences in the intestinal microstructure which could enhance the growth mechanism and body weight. Our findings showed that some microbial components were significantly associated with body weight and their interactions with the host could be inferred to explain why these interactions might alter the host’s metabolic responses. Further investigation into combining bioinformatics with lab experiments in chickens will help us to understand how gut bacteria can change the host’s metabolism by special metabolic features in the gastrointestinal system.

## 1. Introduction

The poultry industry is one of the world’s largest food sectors aiming to boost the producing animal meat at low cost. Body weight is a good sign of overall health and is the primary concern for chicken meat producers [1]. Among other factors, gut health also plays an essential role as a modulatory factor of the production traits. The gastrointestinal tract microbiota is studied widely in poultry production, and several studies have attempted to identify how the intestinal microbes are associated with weight gain and fed conversion ratio traits [2,3]. Some studies focusing on the relationship between gut microbiota and weight gain have suggested its critical role in physiology and gut development, as well as the collection, storage, and expenditure of dietary energy [4,5]. Some investigations focusing on the relationship between gut microbiota and obesity have suggested that gut microbiota affects animal energy utilization and energy deposition [6]. The avian gastrointestinal tract’s histological traits and anatomical function are crucial to its involvement in feed conversion efficiency [7,8]. The intestinal epithelium is a barrier between the external and internal environment of the organism and it plays a significant role in nutrient absorption [9]. The luminal microbiota and the mucosal microbiota are two types of gut microbiota that impact each other and interact with gut health [10,11]. Comparison between the lumen- and the mucosa-associated microorganisms has revealed much greater microbial community richness in the mucosa, particularly in the ileum and caecum of broiler chickens [12]. In the frame of the co-occurrence evaluation between the jejunal content (JC) and mucosa (JM), the two genera (Trichococcus and Oligella) in the jejunal content were found to have a significant positive correlation with other genera of the JM only in low-weight of broiler chickens [13].

In previous studies with broiler chickens, the microbiota analysis focused mainly on the gut of broiler chickens, and attention has rarely been paid to the bacteria of local breeds that are dual-purpose breeds. With global warming, poultry production’s improvement will depend on utilizing and developing local chicken breeds [14] and their gut microbiota. There are few reports in the available literature on integrating bioinformatics with lab experiments in chickens, which are rare in local chicken breeds. Whether the microbiota, gene expressions, and histological characteristics of the gastrointestinal tract are altered by the growth performance in local chickens becomes an interesting question. Most local chickens in Egypt are dual-purpose breeds and the Golden Montazah chicken is one of three local strains with the highest growth, body weight gain, and solid reproductive qualities [15]. The aim of the present study was to investigate the mechanism of growth by comparing the high weight (HW) and low weight (LW) in a local breed of chickens by studying the differences in the gut microbiota, gene expressions, and histological characteristics of their gastrointestinal tracts. Moreover, a multivariable association analysis was used among all of the data (i.e., microbiota and other traits) and body weights of the birds.

Here, we sequenced the V4 region of the 16S rRNA gene to describe the microbiota diversity, components, and predicted functionality to further investigate the differences in the microbial community structure and functional capacity between the high-weight (HW) and low-weight (LW) chickens in the local breed, Golden Montazah. By comparing the abundances of microbial populations between these two groups, we determined whether the presence of certain bacteria was correlated with growth performance.

## 2. Materials and Methods

### 2.1. Bird Management

The protocols were approved by the Animal Production Research Institute’s (APRI) animal care and use committee (ethical approval number: 2023393429). The current research was conducted in the APRI at the poultry research farm in el-Azab, in the Fayoum governorate, Egypt. A total of 480 Golden Montazah chickens were housed in batteries, 20 cages and 24 birds per cage, within a brooder pen with a temperature of 30 °C for three days, and this was then gradually reduced by 3 °C per week until reaching 24 °C. A thermostat automatically controlled the temperature along with manual ventilation control. Throughout the study, all of the birds were fed a mash diet ad libitum and had free access to water. From 1–20 days of age, the birds were fed a starter diet, followed by a grower diet from day 21 to the end of the study. The diet’s composition is shown in Table 1. Nutrient compositions (Table 1) were calculated according to NRC, 1994. For the first three days, the photoperiod program was 24 h (hours), then 20 h until day 7, and then 16 h thereafter. On the first day at the hatchery, all of the birds were vaccinated against Newcastle disease, infectious bronchitis, and Marek’s disease.

### 2.2. Experimental Design and Sample Collection

On the day of hatching, all of the birds were wing-banded and weighed. At 49 days of age, the birds were ranked according to their weight and the 48 birds with the lowest and highest weights were chosen. The difference between the mean body weight gain on the hatching day and the mean of the highest and lowest body weight gains at 49 days was used to calculate the gain weights. Samples were taken from chickens in the high-weight (HW) group and low-weight (LW) group (*n* = 48 chickens included 12 males and 12 females for each group). The chickens were slaughtered, and the blood was collected after withdrawing their feed for 12 h to reduce their gastrointestinal contents. Each bird’s gut was removed shortly after death, and the length of the whole intestine, including the cecum, was measured.

Additionally, the liver, gizzard, breast, and leg muscles were weighed. All adhering anatomical structures (i.e., mesentery, associated blood vessels, fat, and the pancreas) were removed from each organ at the time of collection to prevent stretching of the intestine. The relative length of the intestine was determined as equal to the weight of the liver, gizzard, breast muscle, and leg muscles and was calculated as follows: [intestine length (cm), organ weight/live body weight (g)] × 100. Blood samples from the birds were centrifuged at 1500× *g* for 20 min, and serum samples were frozen in liquid nitrogen and kept at −80 °C for further analysis.

### 2.3. Blood Biochemical Parameters

Blood samples collected from the chickens (*n* = 48 chickens) were used to assay the biochemical parameters. Total protein, albumin, triglycerides (TAG), and glucose were determined biochemically in the collected sera according to the manufacturing instructions of the Biodiagnostic company kits (Dokki, Giza, Egypt; www.bio-diagnostic.com, accessed on 12 March 2020). The total protein and albumin differences in the collected samples were used to determine the globulin levels. The biochemical parameters were measured using a UV-VIS spectrometer (model T60UV, PG Instruments Limited, Lutterworth, UK). Samples were taken from chickens in the high-weight (HW) group and low-weight (LW) group (*n* = 48 chickens, divided into 12 males and 12 females for each group). 

### 2.4. Histological Characteristics

Samples of jejunum were fixed in neutral buffered formalin. Then, the tissues were dehydrated in an ascending graded series of ethanol and embedded in paraffin wax. Serial sections were cut at 5 μm with a microtome (Galileo SEMI, Diapath, Italy). Four cross-sections of jejunum per bird were stained with Mayer’s hematoxylin and eosin (H&E). Histological characteristics in the jejunum included recording histological morphometric parameters and examining ultrathin sections using electron microscopy. According to Alshamy [16], morphometric measures in H&E-stained sections of the jejunum included villus height (VH), epithelium height (EH), crypt depth (CD), tunica muscularis (TM) thickness, and mucosal enlargement factor (EF) (Appendix A). EF = total mucosal surface length of ten adjacent villi divided by the length of the lamina muscularis mucosa was measured in four cross-sections per bird. At Al-Azhar University, histological morphometric characteristics of the jejunum were studied using a Leica light microscope and imaging software from Leica Microsystems Leica (Application Suite 3.1.0 software, Leica, Wetzlar, Germany). Ten villi, ten crypts, and ten tunica muscularis were measured in four cross-sections per bird.

### 2.5. Electron Microscopic Examination

Three high-weight (HW) birds and three low-weight (LW) birds had their jejunum fixed in 3% glutaraldehyde in 0.1 M sodium cacodylate buffer (pH 7.0) for 2 h at room temperature, rinsed in the same buffer, and post-fixed in 1% osmium tetroxide for another 2 h at room temperature. The samples were dehydrated in an ethanol series ranging from 10% to 90% for 15 min in each alcohol dilution, followed by 30 min in absolute ethanol. The samples were eventually penetrated by pure resin through a gradient of epoxy resin and acetone infiltrations. On a Leica Ultracut R ultramicrotome (Leica; Wetzlar, Germany), four ultrathin sections per bird were cut at 0.5 μm. Copper grids were used to capture ultrathin sections. The semi-thin sections were stained twice with uranyl acetate and lead citrate. Al-Azhar University used a transmission electron microscope JEOL JEM 1010 (JEOL, Ltd., Nihon Denshi Kabushiki-Japan) at 70 kV to investigate the stained sections. Microvillus length (μm) and terminal web height (μm) were used to determine the ultrastructure of the jejunum. Forty individual microvilli and twenty loci in the terminal web were measured in the four ultrathin sections for each bird.

### 2.6. Gene Expression 

Peptide transporter 1 (PEPT1), glucose transporter 2 (GLUT2), acetyl-CoA carboxylase (ACC), and carnitine acyl transferase I (CPT1) genes were selected for their roles in the absorption and biosynthesis of nutrients, peptides [17], glucose [18] and fatty acids [19], in the small intestine and for energy production in mitochondria [20], receptively. Also, these genes are in relation with growth performance [9]. To quantify the expression of intestinal nutrient transporter genes, the mucosa from the jejunum of 28 birds (*n* = 14 per group) was collected and frozen in liquid nitrogen before being kept at −80 °C. Total RNA was isolated from the mucosa in the jejunum using the Trans-Zol reagent (Beijing, China), as directed by the manufacturer’s instructions (Lot no: 31206). The cDNA was synthesized from total RNA using the Revert Aid First Strand cDNA Synthesis kit (Thermo Fisher Scientific, Vilnius, Lithuania, Lot no. 01099653) according to the manufacturer’s instructions. Oligonucleotide primer sequences for beta-actin (β-actin), ACC, CPT1, PEPT1, and GLUT2 genes were synthesized by Invitrogen (ThermoFisher Scientifics, Vilnius, Lithuania; Table 2). PCR amplifications were carried out in 25 μL reactions containing 12.5 μL of Maxima SYBR Green qPCR master mix (2X) (ThermoFisher Scientific, Vilnius, Lithuania), 1.0 μL of each primer (10 pmol final concentration), 8.5 μL of water, and 3 μL of cDNA template. The PCRs were carried out in a real-time thermal cycler machine (Rotor-Gene Q, QIAGEN, Hilden, Germany) with the following thermal cycling conditions: reverse transcription at 50 °C for 30 min, followed by reverse transcription at 95 °C for 15 min, and 40 PCR cycles with denaturing at 94 °C for 15 s, annealing at 60 °C for 30 s, and elongation at 72 °C for 15 s. The RT-PCR data, amplification curves and cycle threshold (CT) values were analyzed using Rotor-Gene Q Series Software 2.3.4 (Build 3). According to Yuan et al. [21], relative gene expression in distinct samples was determined by comparing the CT value of each sample to that of the positive control.

### 2.7. Microbial Composition Sequencing and Analysis

The microbial composition study focused on only the jejunum in consideration of the many pieces of literature that have studied the ileum and cecum. In addition, it is the main segment of the small intestine that is used for absorbing and transferring the nutrients that are related to growth performance in chickens.

Samples of the jejunum were taken from the 24 chickens with the highest weight in both content (HC; *n* = 12) and mucosa (HM; *n* = 12), as well as a lowest weight in both content (LC; *n* = 12) and mucosa (LM; *n* = 12) and they were kept at −80 °C. The microbial composition of the high and low body weight chickens was compared using 16S rRNA gene sequencing. Total bacterial genomic DNA was isolated from samples using PrepMan™ Ultra Sample Preparation Reagent DNA extraction kits (Thermo Fisher Scientific, London, UK, and Catalog Numbers 4318930). The Ion 16STM Metagenomics kit (Fisher Scientific, London, UK, catalog Number A26216) was used to sequence the 16S rRNA gene in bacterial genomic DNA. For PCR amplification, an aliquot of each extracted DNA sample was utilized as a template. Additionally, 16S rRNA, library preparation, and DNA sequencing were performed by a commercial provider (Personalbio Co., Ltd., Shanghai, China). The primers targeting the V3–V4 region (forward 5′-CCT AYG GGR BGC ASC AGG NG-3′, reverse 5′-GGA TAC NNG GGT ATC TAA T-3′) were utilized for amplification, and PCR products were purified to produce an estimated amplicon size of 570 bp [22]. The PCR conditions were as follows: initial denaturation, annealing, and extension were carried out and repeated at 94 °C for 4 min, 94 °C for 30 s, 50 °C for 45 s, and 72 °C for 30 s for a total of 25 cycles.

After removing the chimeric sequences (less than 160 bp), quality sequences were recorded for each sample. By using open-reference OTU selection and UCLUST, the sequences were grouped into OTUs with 97% similarity [23]. The Greengenes default database in QIIME was used to identify the taxonomy of the OTUs [24]. The most prevalent OTUs across the groups were then categorized using BLASTN for taxonomy classification, targeting the 16S rRNA marker against the NCBI nucleotide database [25]. 

For the analysis of bacteria and archaea, SILVA (Release 138.1, https://www.arb-silva.de, accessed on 12 January 2022) and RDP (Ribosomal Database Project, Release 11.1, http://rdp.Cme.msu.edu/, accessed on 18 February 2022) databases for the 16S rRNA gene were used by default to identify OTU diversity among the samples and between the groups [26]. The detected taxonomic ranks were presented independently as heat maps and clustered using Euclidean distance.

Using the R program, a Venn graph was created to compute the total number of OTUs per sample (i.e., per group). PICRUSt-2 was used for functional prediction analysis to predict the metabolic functioning of bacteria and archaea [27]. PICRUSt-2 predicted the association function of 16S rRNA gene sequences utilizing three functional profile databases: KEGG, COG, and Rfam. KEGG orthologous gene cluster analysis was applied to discover biological function pathways: metabolism, genetic information processing, environmental information processing, cellular processes, and organismal systems levels and sublevels, and the data were presented as boxplots.

### 2.8. Statistical Analysis

All of the data were represented as means with standard deviations (SEM) and subjected to one-way ANOVA with SAS 2002 software’s GLM technique (SAS, Cary, NC, USA). The individual animal was considered as the experimental unit. The statistical model used was the effect of the weights of the two groups, the high and low weight, and the sex effect. A *t*-test was used to compare the means, and the differences were considered significant at *p* < 0.05. The males and females selected in the highest and lowest birds were equal as shown above, thus the sex effect was not an effective factor to be considered.

Based on the body weight of the sampled birds, the OTUs presented in at least three of the four replicates for each of the six high-weight and/or six low-weight birds were retained for further analysis. All the traits measured for the 12 selected birds were analyzed for association with the filtered microbiota. The multivariable correlation analysis was conducted on orange software using Spearman’s correlation (version 3; available at https://libraries.io/pypi/Orange3/, accessed on 28 April 2022) [28]. 

Before performing the correlation analysis, all of the data (i.e., microbiota and other traits) were transformed using z-score transformation independently and visualized using a correlation distance map. Then, variables contrasting high-weight and low-weight birds were selected, the transformed z-score data of the selected cluster were represented as a heatmap, and both the selected variables (rows) and the samples (columns) were clustered using Euclidean distance. Finally, the formed clades were validated by the bird-clustering-by-weight criterion (i.e., the traits clade was considered when the studied birds were clustered by their weight as high or low).

## 3. Results

### 3.1. Bodyweight and Weight Gain

On day 49, the mean final body weight in high-weight (HW) birds was 2.17 times greater than in the low-weight (LW) birds (Table 3). The daily body weight increase in the high-weight group rose at a rate of 14.5 g per day from day 1 to day 49 after hatching, compared to 6.7 g in the low-weight group. The relationship between carcass characteristics and body weight was not significantly different (Table 3).

### 3.2. Gastrointestinal Traits and Histological Characteristics

The differences in intestinal length and liver weight between the HW and LW birds were significant (*p* < 0.05) when the gut features were compared to the body weight (Table 4). The height of the jejunal villus was 11.4 percent greater in the HW birds than in the LW birds (Table 4). In the HW chickens, the crypt depth of epithelium height and the enlargement factor were substantially larger (*p* < 0.05) than in the LW chickens (Table 4).

### 3.3. Electron Microscopic Examination

Mitochondria (M) of various shapes, including rod-like, oval-shaped, and tadpole-shaped mitochondria, were found in the HW chickens (Figure 1A). As seen in the HW chickens, the goblet (G) cells had narrow bases linked to the basement membrane, and their cell bodies extended into the lumen (Figure 1B). Lysosomes (L) are small, roughly spherical organelles found inside cells, measuring around a millimeter in length (Figure 1A–D). In the LW chickens, there were fewer mitochondria (M) and unbound ribosomes (R), as well as fewer mitochondrial lysis (Figure 1C). The goblet cells’ ovoid nucleus and cup-shaped apical section were found basally in the slender stem-like region (Figure 1B). Additionally, a few unbound ribosomes and lysosomes dispersed across epithelial cells at the luminal surface (Figure 1C,D). The microvilli length and terminal web are shown in Figure 1A,B. On 49 days of age, there was a significant difference (*p* < 0.05) in the microvilli length (µm) in the jejunum of the high-weight group compared to the low-weight group (Table 4). However, the terminal web thickness (µm) did not vary substantially between the high-weight and low-weight groups (Table 4). 

### 3.4. Gene Expression in Jejunum Mucosa 

The ACC, GLUT-2, PEPT-1, and CPT-1 transporter genes’ expression in the jejunal mucosa of the HW and LW chickens did not change substantially (*p > 0.05*; Figure 2). HW chickens had 14.6, 1.7, and 42.4% higher GLUT-2, PEPT-1, and CPT-1 genes levels than the LW chicks (Figure 2). The LW chickens had 26.3% higher ACC mRNA levels than the HW chickens (Figure 2).

### 3.5. Blood Biochemical Parameters 

The high and low weights of the chicken had an impact on the blood biochemical parameters. The findings showed that total protein, albumin, and triglycerides levels in the blood were significantly higher (*p* < 0.05) in the HW chickens than in the LW chickens, while glucose levels were not significantly different (*p* < 0.05). The glucose levels in the HW birds were 17.5% higher than in the LW birds (Table 5).

### 3.6. Microbial Profile

#### 3.6.1. Alpha and Beta Diversities 

The alpha diversity of the bacterial communities demonstrated that HW and LW in content and mucosa had no significant influence (*p* > 0.05) on bacterial community alpha diversity as estimated by Chao1, Faith, Good’s coverage, Shannon, Simpson, Pielou indices in addition to the total observed species (Figure 3). 

In terms of the effect of the birds’ weights and the content and mucosa on beta diversity indicators, the results in Appendix A clearly show that there were no significant differences (*p* > 0.05) in beta diversity indicators, principal component analysis (PCA), nonmetric multidimensional scaling (NMDS), and principal coordinates analysis among the HC, HM, LC, and LM groups (PCoA). However, the comparative examination of intergroup-group differences in weighted UniFrac distance revealed substantial disparities between the HC and HM groups (Appendix A).

#### 3.6.2. Species Composition

The assignment of consensus taxonomy resulted in the identification of 16 phyla in Golden Montazah chickens. The average relative abundance levels of microbiota at the phylum were dominated by Firmicutes (45.8%) followed by Proteobacteria (41.5%), Actinobacteria (2.9%), Bacteroidetes (8.0%), and Verrucomicrobia (0.4%) that constituted 98.2% of the whole identified phyla. A similar microbial picture was mirrored in the jejunum with Firmicutes and Proteobacteria in the content of the high-weight (HW) and low-weight (LW) group and also in the mucosa of the HW and LW group (Appendix A). The average relative abundance of Actinobacteria phylum was 5.0 in mucosa the HW versus only 0.72 in the LW (Appendix A). The Bacilli were the dominant 16S rRNA sequences in jejunum libraries, followed by the Comamonadaceae and Lactobacillaceae family (Appendix A). The Planococcaceae family was identified in the mucosa of high-weight chickens with the highest values (Appendix A).

At different taxonomy levels, correlations among microbes in Golden Montazah chicks are shown in Figure 4. At the phylum level, Actinobacteria was highly abundant in the mucosa of high-weight birds (HM), while the Firmicutes and Deferribacteres were equally abundant in the mucosa of high and low-weight birds (Figure 4). Lentisphaerae was highly abundant in the mucosa of low-weight birds (LM). Families that include potentially pathogenic species, such as Clostridiaceae, Bacillaceae, Enterobacteriaceae, Moraxellaceae, and Paenibacillaceae, exhibited a high occurrence in the mucosa of the low-weight birds (LM). Microbacteriaceae and Planococcaceae were exclusively abundant in the mucosa of the high-weight birds (HM) in contrast to Comamonadaceae and Flavobacteriaceae, which were solely found in the content of the high-weight birds (HC). Caulobacteraceae and Bifidobacteriaceae families were abundant in the content of high and low-weight birds. Lactobacillaceae S24-7 was found in all groups but was much higher in the high-weight birds than in the low-weight birds regardless of the sampling type (HC and HM; Figure 4).

#### 3.6.3. Microbial Function Prediction

The findings of PICRUST’s functional prediction analysis of the microbiological activities based on the KEGG pathway among the HC, HM, LC, and LM groups are shown in Figure 5. Amino acid, cofactor, and vitamin production were considerably (*p* < 0.05) higher in the HC group compared to the HM, LC, and LM groups in terms of biological biosynthesis pathways (Figure 5A). In comparison to the high growth performance of the HC and HM groups, the microbial degradation routes of inorganic nutrients, carboxylate, and carbohydrates were elevated (*p* < 0.05) in the lower growth performance of the LC and LM groups (Figure 5B). In comparison to the HM, LC, and LM groups, the HC group had the greatest (*p* < 0.05) microbial breakdown pathways of nucleic acid processing, glycolysis, and cellular metabolite energy (Figure 5C).

#### 3.6.4. Multivariable Correlation

The correlations between the measured parameters of different variables, including the detected microbiota of the high and low weigh birds, were tested by Spearman’s correlations. Several correlation blocks were detected; however, the correlation block including the body weight trait was selected (Figure 6A). The selected correlated variables were represented as a heatmap using the transformed z-score data to confirm their cladistics relationship using the Euclidean distance. The selected variables were able to distinguish the high-weight birds from the low-weight ones (Figure 6B). From the variables’ perspective, the body weight and the weight gain were highly correlated with Microbacterium, and Sphingobacteriales and lowly correlated with unclassified Enterobacteriaceae, Clostridiales, unidentified Propionibacterium and Lachnospiraceae, Candidatus Arthromitus, and also with gene expression of ACC and CPT-1, and gizzard weight and cecum length (Figure 6B).

## 4. Discussion

Comparing the microbiota, histological characteristics, and gene expression in the highest weight (HW) and the lowest weight (LW) chickens of the Golden Montazah breed showed that the blood biochemical, relative intestinal length, and histological morphometric parameters were significantly different (*p* < 0.05). In the LW chickens, there were fewer mitochondria and ribosomes (R) as shown in electron microscopy examinations. The alpha diversity of microbial composition did not significantly differ between the HW and the LW chickens, but the Actinobacteria phylum was higher in the mucosa of the HW chickens than the LW chickens. Some of the predicted analyses of microbial functions were significantly associated with the mucosa of the HW chickens, which may have improved the intestinal microstructure and its functions. According to Spearman’s correlations, there is a positive correlation between the body weight and the weight gain with Microbacterium and Sphingobacteriales. 

The Golden Montazah chicken is a dual-purpose breed that is one of three local strains with optimum production (growth and body weight) and reproductive traits Youssef et al. [15]. The Golden Montazah chickens were significant different in individual body weight although they were fed the same feed. This could be attributed to the genetic variance component in local chicken breeds being higher than those in foreign breeds [29]. Overall, the body weight and weight gain values of Golden Montazah chickens are close to those reported by Youssef et al. [15], which were 770 (g) and 680 (g) at eight weeks. The increasing relative gizzard weight and intestinal length in the HW chickens increased the degree of food processing and increased starch availability in the gut, which is linked to improved peristaltic movement and higher nutritional concentrations [16]. In the HW birds, the muscle layer regulating gut motility and the tunica muscularis of the intestinal tunica may promote contact between the mucosa and the intestinal content, which might alter absorption processes [7]. The migration of proliferating crypt cells up to the villi ensures continuous regeneration of the small intestinal epithelium [30], which might improve the epithelium height and mucosal expansion factor in the current investigation. This might result from enhanced nutrient absorption and animal growth [31]. In the HW chickens, examination of the ultrastructure in the jejunum revealed an increase in mitochondria and ribosomes in the epithelial cells of the jejunum. Mitochondria are vital for creating ATP and managing cell death [32,33]. Ribosomes are where an mRNA molecule’s nucleotide sequence is translated into protein in the cytoplasm. These differences may be seen in the HW birds’ longer microvilli and terminal web compared to the LW birds. Secretion, mechanotransduction, absorption, and cellular adhesion are all functions of the microvilli on the surface of absorptive cells, as stated by Yamashiro [34]. Reduced goblet cells and lysosomes in the LW group may have contributed to the current study’s reduced epithelium height and mucosal expansion factor. Goblet cells, however, are thought to protect the mucosal membrane of the gut by producing and secreting a variety of mediators, including the mucin MUC2 [35], while lysosomes serve as the cell’s digestive system [36]. Specific transporters present in the brush border of the small intestine convey the building blocks of proteins, lipids, and carbohydrates for absorption by enterocytes [31]. The histological characteristics in the jejunum were studied in three high and three low weights and these are preliminary results, so there is need to a larger sample size to confirm these results.

Improving food transport capacity and regulating the transporter-producing genes increases the entry of nutrients into the intestinal epithelial cells and eventually into the body [36]. In the HW birds, the expression of the intestinal GLUT-2 gene was upregulated, allowing glucose, fructose, galactose, and mannose to enter via the intestinal basement membrane and into the liver, where they are transformed into glucose and subsequently distributed throughout the body [30]. In addition, increased GLUT-2 gene expression might indicate an improved absorption capacity in birds [37]. In the HW birds, PEPT-1 and CPT-1 gene expression were upregulated in HW. Enterocytes are aided in their absorption of di- and tripeptides from the lumen by the PEPT-1 gene [38]. CPT-1 is a mitochondrial enzyme that is involved in the synthesis of acylcarnitine, which is then transported to the mitochondrial matrix for β-oxidation energy generation, with the energy being stored as ATP [19,39]. The lysis of mitochondrial contents in the jejunum of LW might be explained by lowering CPT-1 gene expression in LW. The difference in blood total protein and albumin levels between HW and LW after feeding on the same meal might be related to greater protein consumption to meet the demand for a larger body [40]. Globulin levels in HW and LW serum may have resulted in the same health condition for all of the birds. The glucose level is a significant physiologic aspect under persistent tight regulation [41]. Blood glucose levels are impacted by carbohydrate ingestion, and the primary regulatory mechanism occurs via glucose transporter protein types [42] such as GLUT-2, which was not significantly different between the HW and LW groups. A slight difference between the two chicken groups might also be related to the HW group’s quick increase in skeletal muscle mass [43]. Fasting before slaughtering chicken significantly increased triglyceride levels depending on body weight [44], which was high in the HW group.

Among the observed OTU numbers, the Chao1 and observed species indices tended to show greater diversity across the groups, but microbial α and β diversities did not differ significantly. Our results are similar to those of Liu et al. [45], who also failed to detect differences in the diversity of the ileum and cecal microbial community between chickens with different feed efficiencies [45]. In a recent comparison between the microbiota composition in two body weight groups of a commercial line (i.e., Ross), no significant differences were found in the alpha diversity at the jejunum contents—JC, jejunum mucosa—JM, and caecum contents—CC places, but β-diversities showed considerable separation between the JC, JM, and CC [13]. In our study, similar results were found, which suggested that the gut microbiota within the studied chicken was identical and exhibited limited diversity [13]. The gut microbiota composition at the phylum level in Golden Montazah chickens was dominated by Firmicutes, Proteobacteria, Bacteroidetes, and Actinobacteria, a domination that has been previously reported in similar studies (e.g., [22,46,47]. Values the relative abundance at the phylum level are close to those reported in the indigenous Indian chickens, Aseel, that were Bacteroidetes (44%), Firmicutes (43%), Proteobacteria (6%), and Actinobacteria (1%), for the whole phyla [48]. Additionally, the small numbers of Actinobacteria in the local Egyptian breed, Golden Montazah chickens, were observed previously in Omani chickens [49] and indigenous Indians Aseel chickens [48]. In the jejunum, a high number of Actinobacteria was noticed in high-weight (HW) birds on day 49. Actinobacteria is important in the upper digestive tract of high-producing birds such as Ross 308 [50] and Cobb 500 [49] chicken strains. This indicates that there are profound differences in the microbial population in the intestinal observed between the native and commercial breeds. The increase in the phylum Proteobacteria, which includes many potentially pathogenic bacteria, correlates with a pro-inflammatory cytokine profile. In contrast, the increase in members of the phylum Firmicutes is associated with an anti-inflammatory state and an imbalance in the intestinal microbiota [51] that is observed in the local breeds. In the present study, the Bacilli were the dominant 16S rRNA sequences jejunum libraries which is in harmony with the findings Al-Marzooqi et al. [49]. Increasing the Lactobacillaceae in high-weight chickens is compelling and in connection with reports of Singh et al. [3] and Siegerstetter et al. [52], who reported that Lactobacillaceae in microbial taxa are associated with high productivity in the guts of chickens. Actinobacteria and Lactobacillus showed critical roles in weight gain, and they were shown to be considerably higher with increasing weight gain. Similar to our findings reported by Farkas et al., [13], who stated that the co-occurrence interaction results in the jejunum revealed a correlation between the genera of Actinobacteria and Firmicutes Bacilli classes with different patterns in the two BW groups. Actinobacteria are the highest producers of bioactive secondary metabolites and can be used as a probiotic candidate, particularly in the poultry industry [53]. According to the previous report, Lactobacillus can trigger the body to produce immune globulin and enhance host immunity to gastrointestinal infections [54]. The abundance of the Planococcaceae family increased in the present study; this family was previously reported to be highly co-abundant with the Lactobacillaceae family [55].

In this study, PICRUSt-2 was used to predict the contents and mucosa of the intestine of low-weight (LC and LM) and high-weight (HC and HM) chickens, and many distinct KEGG pathways were detected in the high and low-weight birds. Three KEGG biological biosynthesis pathway (amino acid, cofactor, and vitamin biosynthesis)- linked genes were associated with high-weight activities, and the functional cofactor biosynthesis was found to improve the growth performance of high-weight (HC and HM) chickens considerably. This shows that the gut dynamic microbial community of high-weight (HC and HM) chickens’ gut bacteria might be used as a probiotic for growth promotion [56]. In contrast to the low-weight (LC and LM) groups, the gut microbiota of the HC and HM groups can digest complex and simple carbohydrates and produce more nutrients, such as vitamins, microbial proteins, and volatile fatty acids, allowing the host to gain weight. However, inorganic nutrient breakdown processes, such as carboxylate and carbohydrate, were more prevalent in the low-weight (LC and LM) group than in the high weight (HC and HM) group. These pathways are linked to the host catabolism of muscle responses. Microbiota is vital to the functioning of some microbial components and their interactions with a host [57]. Therefore, it can be inferred that host–microbe interactions exist and our research explains why these interactions might alter the host’s metabolic responses. The predicted analyses of microbial functions, amino acid, cofactor, and vitamin biosynthesis were significantly associated with increased weight gain in chickens. This could enhance the host anabolism by increasing free ribosomes, lysosomes, and abundant mitochondria, which could promote gastrointestinal functions through mucosal enlargement with longer microvilli and terminal web in absorptive cells in the jejunum. This result is supported by the findings of Heinken and Thiele [58], who reported that metabolites synthesized by the microbiota include compounds that can directly regulate and modulate host metabolisms such as neurotransmitters and hormones. This is in harmony with the suggestion that gut microbiota can be considered as a different endocrine organ [59].

In Spearman’s correlations between body weight, the studied parameters, and the counted microbiota, there is a positive correlation between the body weight and the weight gain with Microbacterium and Sphingobacteriales. These results are in harmony with the findings of Zhang et al. [60] who reported Microbacterium and Sphingomonas related to lipid metabolism were observed at a significant level across all of the chickens thus suggesting a significant contribution to the development of gut microbiota for chicken growth. Moreover, the clusters of orthologous groups (COG) analysis revealed 20 lipid metabolism genes associated with Microbacterium [61]. Moreover, we found a low correlation between body weight with unclassified Enterobacteriaceae, Clostridiales Lachnospiraceae and, Candidatus Arthromitus. The microbial diversity in the ilea decreased and overrepresentation of Enterobacteriaceae and underrepresentation of Clostridia was found in the chickens infected with a virus [62].

## 5. Conclusions

The mechanism of growth was enhanced by increasing some organelles of the cell, mitochondria and free ribosomes, which could promote gastrointestinal functions in the jejunum. This could be attributed to the host genetics that have a high genetic variance component in local chicken. The microbiome profile in the gut showed a profound difference between the local and commercial breeds which included Microbacterium and Sphingomonas that were positively correlated between the body weights. Through studying the host–microbe interactions, our research cleared the host–microbe metabolomics that appeared in increasing free ribosomes, lysosomes, and abundant the mitochondria, which could promote gastrointestinal functions. These results could support the hypothesis that the metabolites synthesized by the microbiota include compounds that can directly regulate and modulate host metabolisms.

## Figures and Tables

**Figure 1 animals-12-03474-f001:**
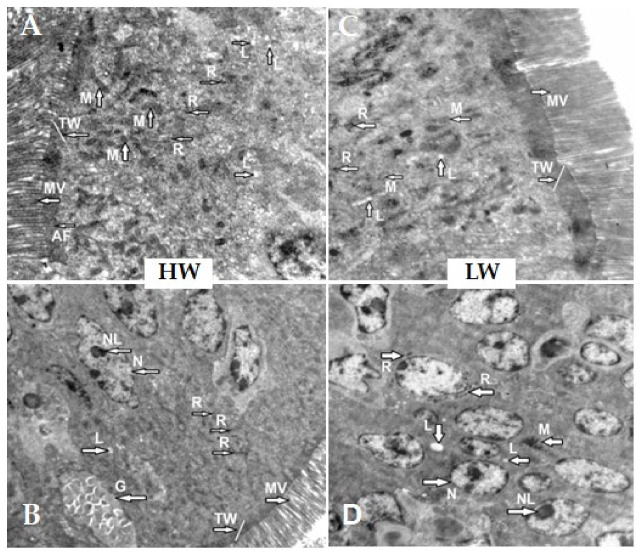
(**A**–**D**). Transmission electron microscopy of the jejunum of high-weight and low-weight birds. Mitochondria (M); lysosomes (L); ribosomes (R); microvilli (MV); goblet cell (G); nucleus (N); lysosomes (L); and nucleolus (NL). Bar: 500 nm.

**Figure 2 animals-12-03474-f002:**
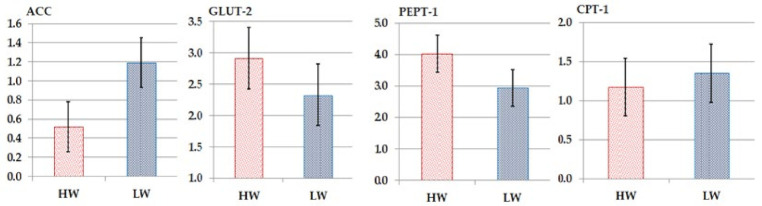
The fold change of the expression of the nutrient transporter genes in the jejunum of high-weight (HW) and low-weight (LW) chickens: acetyl-CoA carboxylase (ACC), carnitine acyltransferase I (CPT1), peptide transporter 1 (PEPT1) and glucose transporter 2 (GLUT2).

**Figure 3 animals-12-03474-f003:**
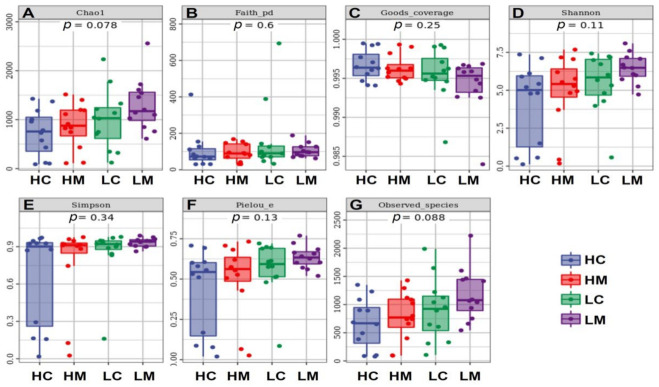
Alpha diversity measures of bacterial communities in jejunum with high weight in content (HC) and mucosa (HM) and low weight in content (LC) and mucosa (LM). (**A**): Chao1, (**B**): Faith, (**C**): Good’s coverage, (**D**): Shannon, (**E**): Simpson, (**F**): Pielou indices, and (**G**): observed species.

**Figure 4 animals-12-03474-f004:**
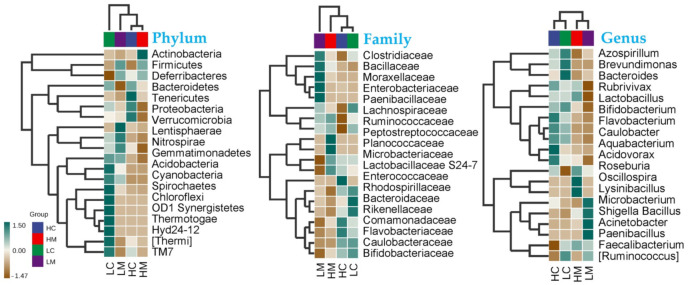
Heat map for total species abundance (transformed by z-score) among high and low-weight birds in both content (C) and mucosa (M) gut microbiota at different taxonomical levels.

**Figure 5 animals-12-03474-f005:**
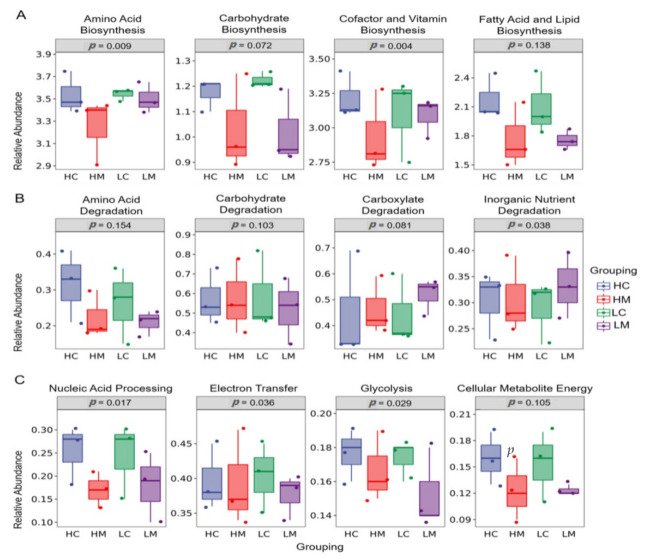
Phylogenetic investigation of communities by reconstruction of unobserved states (PICRUST) analysis of microbial functions based on the KEGG pathway among groups of high weight in both content (HC) and mucosa (HM), as well as low weight in both content (LC) and mucosa (LM) is shown in (**A**) biosynthesis pathways, (**B**) degradation pathways (**C**) Nucleic acid processing, glycolysis, and cellular metabolite energy (*p* < 0.05).

**Figure 6 animals-12-03474-f006:**
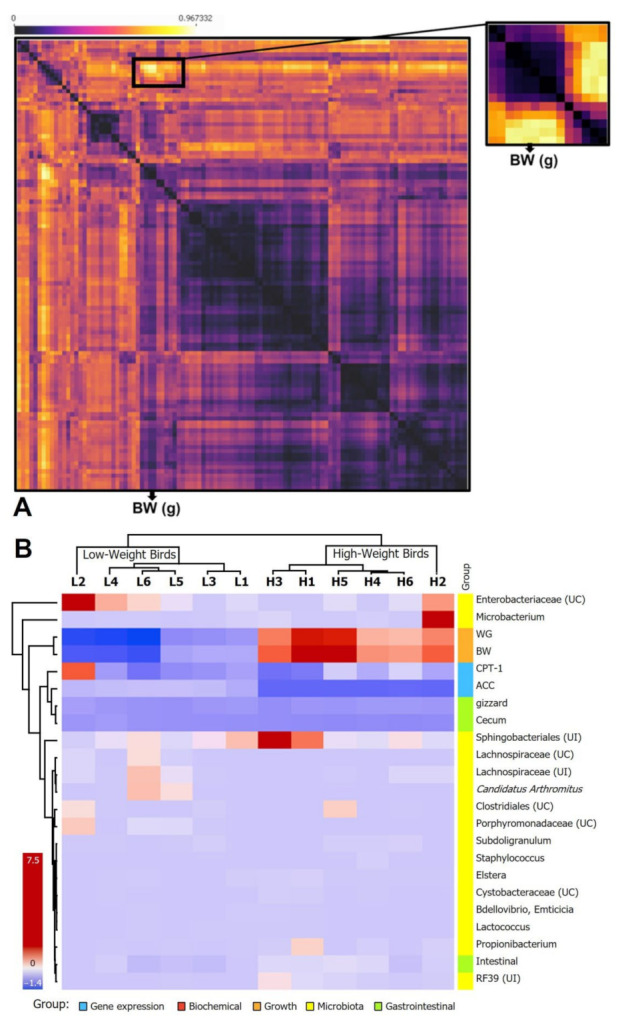
(**A**): Heatmap of Spearman’s correlations between body weight, the studied parameters and the counted microbiota. (**B**): Heatmap of the correlations between body weight, the studied parameters and the microbial composition and the colors range from black (low correlation) to yellow (negative correlation).

**Table 1 animals-12-03474-t001:** Starter and growing experimental diet composition (%) and nutrient composition.

Components	Starter	Grower
Ingredient (%)		
Yellow corn	62.49	71.08
Soybean meal (CP 44%)	22.59	13.17
Corn gluten meal	10	10
Soya oil	0.4	1.27
L-lysine HCl	0.51	0.51
DL-methionine	0.12	0.09
Limestone	1.3	1.22
Dicalcium phosphate	1.76	1.8
Salt	0.33	0.36
Choline chloride 60%	0.1	0.1
Sodium bicarbonate	0.1	0.1
Premix ^1^	0.3	0.3
Total	100.00	100.00
Nutrient composition ^2^ (calculated analysis)		
Metabolizable energy (kcal/kg)	3025	3175
Crude protein, %	21.5	18
Dry matter	91.4	91.9
Crude fiber	3.18	2.61
Lysine %	1.3	1.05
Methionine + Cysteine, %	0.95	0.82
Calcium %	0.95	0.95
Available phosphorus %	0.45	0.45

^1^ Provided the following per kilogram of diet: vitamin A, 6000 IU; vitamin D3, 500 IU; vitamin E, 20 IU; vitamin K3, 0.50 mg; vitamin B1, 2.1 mg; vitamin B2, 3.0 mg; vitamin B6, 3.5 mg; vitamin B12, 0.01 mg; pantothenic acid, 10 mg; niacin, 15 mg; biotin, 0.15 mg; folic acid, 0.45 mg; choline chloride, 500 mg; Fe, 80 mg; Cu, 7 mg; Mn, 60 mg; Zn, 65 mg; I, 0.35 mg; and Se, 0.23 mg. The premix was manufactured by the Agri-Vet company, Cairo, Egypt. ^2^ Nutrient compositions were calculated according to NRC, 1994.

**Table 2 animals-12-03474-t002:** Target genes, primer sequences, accession number, and product size in RT-PCR reactions.

Target Gene	Primer Sequences	Accession No.	Product Size (bp)
Carnitine acyltransferase I *(CPT-1)*	F: GACGTCGATTTCTGCTGCT	AY675193	337
	R:GCAGCGCGATCTGAATGAAG		
Acetyl-CoA carboxylase *(ACC)*	F: AATGGCAGCTTTGGAGGTGT	NM_205505	119
	R: TCTGTTTGGGTGGGAGGTG		
Peptide transporter 1 *(PEPT1)*	F:CCCCTGAGGAGGATCACTGTT	NM_204365	205
	R: CAAAAGAGCAGCAGCAACGA		
Glucose transporter 2 *(GLUT2)*	F: CACACTATGGGCGCATGCT	NM_207178	116
	R:ATTGTCCCTGGAGGTGTTGGTG		
βeta-actin	F:CCACCGCAAATGCTTCTAAAC	NM205518	175
	R:AAGACTGCTGCTGACACCTTC		

**Table 3 animals-12-03474-t003:** Body weight, weight gain, and carcass traits in chickens from the high (HW) and low (LW) weight groups.

Variables	HW	LW	SEM	*p* Value
**Body weight and weight gain**				
Body weight at 49 days (g)	760.5	367.0	11.6	<0.0001
Body weight gain (g)	710.9	328.0	11.1	<0.0001
**Carcass traits**				
Relative carcass weight %	56.2	54.6	0.7	0.222
Relative breast muscle weight %	9.9	7.9	0.7	0.156
Relative leg muscle weight %	12.3	11.5	1.2	0.803

HW: high-weight chicken; LW: low-weight chicken; and SEM: standard error of means.

**Table 4 animals-12-03474-t004:** The gastrointestinal traits and histological morphometric parameters in the jejunum of chickens from the high (HW) and low (LW) weight groups.

Variables	HW	LW	SEM	*p* Value
**Gastrointestinal parameters**				
Relative intestinal length (%)	11.0	9.3	0.16	0.006
Relative cecum length (%)	2.5	2.7	0.35	0.220
Relative liver weight (%)	3.1	2.1	0.13	0.005
Relative gizzard weight (%)	3.3	3.0	0.21	0.453
**Histological morphometric parameters in jejunum**				
Villus height (μm)	587.3	527.4	32.21	0.173
Crypt depth (μm)	138.1	116.2	6.53	0.027
Epithelium height(μm)	51.2	43.9	1.83	0.009
Enlargement factor	3.03	1.7	0.21	0.004
Thickness of the tunica muscularis	186.5	161.1	28.81	0.569
**Measurements of ultrastructure in jejunum**				
Microvilli length (μm)	1.1	0.8	0.05	0.001
Terminal web (μm)	0.3	0.3	0.01	0.205

HW: high-weight chicken; LW: low-weight chicken; and SEM: standard error of means.

**Table 5 animals-12-03474-t005:** Serum biochemical parameters in the high (HW) and low (LW) weight chicken groups.

Variables	HW	LW	SEM	*p* Value
Total protein (g/dL)	5.0	4.4	0.12	0.019
Albumin (g/dL)	3.4	2.8	0.09	0.009
Globulin (g/dL)	1.6	1.5	0.07	0.586
Glucose (mg/dL)	215.3	183.2	17.81	0.316
Triglycerides (mg/dL)	173.4	126.7	8.43	0.011
GOT (U/mL)	25.4	30.4	3.34	0.289
GPT (U/mL)	8.8	9.4	1.83	0.673

HW: high-weight chicken; LW: low-weight chicken, SEM: standard error of means, GOT: glutamate-oxaloacetate transaminase, and GPT: glutamate-pyruvate transaminase.

## Data Availability

All raw data from microbial genome sequencing have been uploaded to the National Center for Biotechnology Information (NCBI) and can be found under the Bio-Project number PRJNA785276 (https://www.ncbi.nlm.nih.gov/bioproject/PRJNA785276/, accessed on 1 December 2021).

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
