# Peer review of "Gut Microbiota, Intestinal Morphometric Characteristics, and Gene Expression in Relation to the Growth Performance of Chickens"

_animals, 2022, doi:10.3390/ani12243474_

Round 1
Reviewer 1 Report
Abdel-Kafy and colleagues compared microbiota composition, intestinal histological characteristics, and gene expression in chickens of varying growth performance. The manuscript is well-written and executed. However, I have a minor concern about this study.
Minor issues:
Discussion:
1. Logically, the heaviest chicken indicates good nutrient absorption, which could be attributed to its improved intestinal microstructure and gut microflora. This will result in improved chicken growth performance. In this work, the authors found a significant improvement of intestinal microstructure and gut microbial community in HW chickens. I think that it is a logical result and I don’t see a clear objective in this work. Perhaps the authors could include some explanation in the discussion section to highlight the significance of this study.
2. Perhaps the authors could provide a brief explanation for why there is such a significant difference in body weight between chickens fed the same feed.
3. In this study, the sample for microbiota analysis was taken from the jejunum. Is there any reason why this segment was chosen over another? The results of the microorganism community may differ between GIT segments.
Reference:
Please use proper referencing format. Line 391 and line 407. Please check and correct all.
Overall, this manuscript is well-written and suitable for publication in this journal.
Reviewer 2 Report
The methods used to evaluate the selected breed of chickens described in the Manuscript are valuable and noteworthy. However, my doubts are raised by the use of one breed of hens, which is characteristic for a given area of occurrence, and the generalization of the obtained observations. In addition, I think that the description of Simple Summary should be improved, the Authors needlessly provide specific results in this section. The Abstract does not provide information on the feeding of chickens. Besides, did the authors, when writing "triacylglycerol" in Abstract and elsewhere in the Manuscript, mean "triglycerides"?
The lack of a clear description of the research hypothesis and the aim of the study is a major oversight on the part of the Authors. Please complete this. Moreover, the statistical description should be removed from the introduction (89-92).
In Table 1, there is no description of footnote 2. There is also no information regarding the description of the nutrients of the mixture. Were they analyzed or calculated?
Please describe the basic histological procedure in more detail. Why was an electron microscope used for basic histology? Please describe in detail in both of the histological methods to what thickness were the slices cut, how many repetitions were prepared? How many measurements were taken? Please remove the sentence from lines 140-141 as it appears repeatedly in the text.
In the case of Table 3, I wonder about the sense of comparing these two groups with each other, because it can be assumed that these differences will be statistically significant.
Why was sex not included in the statistical calculations as an experimental factor, since the tests were performed on birds of both sexes?
Conclusions are too general, please correct.
Please correct the References in line with the guidelines for Authors.
Round 2
Reviewer 2 Report
Although the Authors responded to most of the comments in the Manuscript, some were left without any reaction. Besides, I cannot find a letter that would contain replies to my review.
Below are some comments that were not included in the Manuscript:
“The Abstract does not provide information on the feeding of chickens. Besides, did the authors, when writing "triacylglycerol" in Abstract and elsewhere in the Manuscript, mean "triglycerides"?”.
“The lack of a clear description of the research hypothesis and the aim of the study is a major oversight on the part of the Authors. Please complete this.”
“Why was sex not included in the statistical calculations as an experimental factor, since the tests were performed on birds of both sexes?”
The response is: “Sex effect did not present in this study”. It is not possible to exclude gender in such an experimental structure. Authors should correct the calculations if they want to draw correct conclusions.
Too bad that the basic feed analysis has not been performed, but only calculated. Please complete the table with the value of dry matter and fiber.
Still, the Authors did not take into account the comments on the correct preparation of the References, in accordance with the instructions for Authors.